# Understanding the Impact of Personal Resources on Emotional Exhaustion Among Emergency Healthcare Workers: A Structural Equation Modeling Approach

**DOI:** 10.3390/healthcare13182336

**Published:** 2025-09-17

**Authors:** Alicia Arenas, Valeria Terán-Tinedo, Esther Cuadrado, Rosario Castillo-Mayén, Bárbara Luque, Carmen Tabernero

**Affiliations:** 1Department of Social Psychology, University of Seville, 41018 Sevilla, Spain; vteran@us.es; 2Department of Psychology, University of Cordoba, 14071 Cordoba, Spain; esther.cuadrado@uco.es (E.C.); rcmayen@uco.es (R.C.-M.); bluque@uco.es (B.L.); 3Department of Social Psychology and Anthropology, University of Salamanca, 37007 Salamanca, Spain; carmen.tabernero@usal.es

**Keywords:** emergency healthcare workers, job demands-resources model, structural equation modeling, burnout, personal resources

## Abstract

**Background/Objectives**: The COVID-19 pandemic was associated with an increase in burnout vulnerability in healthcare workers, which often translated into higher potential risks for their personal safety and that of their patients. This study was based on the JD-R model and explored emotional exhaustion and stress, as a result of a lack of personal resources such as self-efficacy to cope with stress and poor emotion regulation, and work demands, with the buffering effect of a healthy lifestyle. **Methods**: The sample was composed of 189 emergency healthcare workers from Spain, aged between 26 and 59 years old. A cross-sectional and correlational study was carried out through an online platform. **Results**: The mediation, moderation, and Structural Equation Modeling analyses supported the proposed model, demonstrating good fit to the data. Work demands showed both a positive direct effect and an indirect effect on emotional exhaustion through emotion regulation and stress. In turn, self-efficacy to cope with stress and healthy lifestyle showed indirect effects on emotional exhaustion, also through emotion regulation and stress. Further, healthy lifestyle moderated the relationship between stress and emotional exhaustion, by mitigating the negative influence of stress on emotional exhaustion. **Conclusions**: The study highlights the importance of incorporating self-care in coping with stress and preventing burnout and should be considered when designing interventions in the context of emergency healthcare to improve well-being in its workers.

## 1. Introduction

The COVID-19 pandemic posed an unprecedented challenge to societies worldwide, exposing healthcare professionals to heightened risks to both their physical safety and mental health [1]. During this period, elevated rates of anxiety and post-traumatic stress disorder were reported among medical staff, particularly among female nurses and physicians in China [2]. Comparable findings emerged in the United States, where more than half of healthcare workers experienced mild psychiatric symptoms and over 40% reported signs of emotional disorders [3]. Similarly, across eight European countries, nearly one-third of health professionals described moderate to extremely severe distress [4].

Healthcare workers (HCWs) are especially vulnerable to stress given the life-or-death consequences of their actions and their responsibility for human lives [5]. Those providing direct patient care under crisis conditions face intensified workloads and patient demands in resource-constrained environments, exacerbating psychological strain [6]. Evidence consistently shows that healthcare and emergency personnel are exposed to significant stressors that may manifest as psychosomatic complaints, sleep disturbances, fatigue, irritability, nervousness, and reduced concentration [7]. Chronic stress is further linked to depression [8], occupational injuries [9], accidents, and work-related illnesses [10]. Physician burnout not only impairs individual well-being but also undermines the healthcare system and patient outcomes, producing adverse effects such as lower patient satisfaction, depression, poor self-care, and diminished productivity [11]. Both the WHO and ILO have emphasized that inadequate care and the risk of medical errors are direct threats to patient safety [12].

Burnout, a maladaptive response to prolonged occupational stress, is typically characterized by emotional exhaustion, depersonalization, and reduced personal accomplishment [13]. Emotional exhaustion is considered the central dimension and often precedes the others [14,15,16,17]. Research indicates that emergency medical service (EMS) professionals experience particularly high levels of exhaustion and depersonalization, alongside reduced accomplishment [18]. Gender differences also shape burnout: women tend to show greater emotional exhaustion, whereas men report higher depersonalization [19]. Occupational stress is generally more pronounced in women, often attributed to the dual burden of work and family responsibilities and the lack of time for personal activities [20].

The Job Demands–Resources (JD-R) model [21] has become a leading framework to explain the interplay of workplace demands, resources, and employee well-being in healthcare. Its central premise—that job characteristics can be categorized as demands (e.g., workload, emotional strain) or resources (e.g., autonomy, social support)—has proven highly relevant in health services. The JD-R model identifies two key pathways: (a) a health-impairment process, where excessive demands such as time pressure, emotional burden, and role conflict predict burnout and absenteeism, and (b) a motivational process, where resources such as leadership, team cohesion, autonomy, and fair rewards foster engagement and care quality [22]. Yet, demands are often structural and pervasive—work overload, insufficient staffing, and emotional labor remain widespread stressors [23,24,25]. Moreover, resources vary across roles and contexts: physicians may benefit from autonomy, whereas care assistants often lack access to such protective factors [23]. Public versus private healthcare systems also differ in resource availability, influencing stress outcomes [24]. Additionally, resources are relational as well as structural: qualitative studies show that teamwork and leadership support are unevenly distributed, leaving some professional groups excluded [23].

Beyond organizational conditions, personal resources such as self-efficacy, optimism, resilience, self-esteem, and emotion regulation have also been identified as critical protective factors [26]. These attributes function similarly to job resources: they motivate, buffer stress, and generate positive feedback loops. In healthcare, resilience and self-efficacy are frequently targeted in interventions to help staff manage emotional and workload demands. Empirical evidence shows that personal resources are reciprocally linked to job resources and attenuate the negative impact of high demands on well-being [27,28]. In particular, self-efficacy—the belief in one’s capacity to cope with job-specific challenges—has been associated with reduced burnout and turnover among nurses [29] and was identified as a key predictor of burnout during the COVID-19 pandemic [30]. Since domain-specific self-efficacy better predicts behavior than general self-efficacy [31], examining stress-related self-efficacy among emergency healthcare workers is especially relevant.

Emotion regulation, defined as the ability to modulate emotional arousal to meet situational demands [32,33], has also been linked to psychological stress [30]. Strategies such as mindfulness have demonstrated effectiveness in reducing burnout among physicians [34,35,36]. Moreover, variations in emotion regulation explain why healthcare residents with comparable workloads experience different levels of burnout [37]. Research confirms that stronger self-regulatory skills predict lower emotional exhaustion and depersonalization [38,39] and can mediate the link between job demands and emotional exhaustion [40].

However, growing evidence questions the assumption that resources alone can buffer the detrimental effects of excessive demands, underscoring the need to explore additional protective factors [23]. One promising area concerns lifestyle habits. Healthy behaviors such as regular exercise, balanced nutrition, and adequate rest have been shown to mitigate stress and improve resilience. During the pandemic, reduced physical activity and sleep deprivation predicted anxiety and depression among healthcare workers in China [41]. Similarly, lifestyle factors have been highlighted as crucial moderators in stress–burnout relationships [42,43,44].

Integrating lifestyle habits into the JD-R framework extends its explanatory scope by incorporating recovery-enhancing behaviors beyond traditional psychological or organizational resources [22,26,45]. In healthcare contexts characterized by persistent emotional and cognitive demands, lifestyle factors may function as external moderators that complement personal resources, sustaining energy levels and work capacity. Such integration has practical implications for safeguarding both worker well-being and patient safety [46].

Taken together, this evidence highlights the urgency of assessing psychosocial risks in highly demanding healthcare sectors and of identifying protective factors that may reduce emotional exhaustion. Accordingly, this study applies the JD-R model to examine how job demands, self-efficacy, emotion regulation, and lifestyle habits interact in predicting emotional exhaustion among emergency healthcare workers. Considering the reviewed literature, the hypotheses below are proposed.

As works demands are associated with emotion regulation [40] and to emotional exhaustion [21]; as emotion regulation is related to both stress [33] and emotional exhaustion [40]; and as stress is related to emotional exhaustion [18], it is hypothesized that

**H1:** *Emotion regulation and stress act as serial mediators in the relationship established between work demands and emotional exhaustion*.

Moreover, as self-efficacy to cope with stress is related to emotion regulation, stress, and burnout [29,30], it is hypothesized that

**H2:** *Emotion regulation and stress act as serial mediators in the relationship established between self-efficacy to cope with stress and emotional exhaustion*.

Additionally, as healthy lifestyle has been shown be associated with emotion regulation, stress, and burnout [38,39,42], it is hypothesized that:

**H3:** *Emotion regulation and stress act as serial mediators in the relationship established between healthy lifestyle and emotional exhaustion*.

Furthermore, as healthy lifestyle has been shown to moderate the stress–burnout relationship [44], it is hypothesized that

**H4:** *Healthy lifestyle will moderate the strength of the relationship between stress and emotional exhaustion*.

In brief, we hypothesize the predictive model of emotional exhaustion presented in Figure 1.

## 2. Materials and Methods

### 2.1. Participants and Procedure

The sample was composed of 189 Spanish emergency healthcare professionals working in an emergency service in the south of Spain, of which 49.7% were male and 50.3% were female, aged between 26 and 59 (M = 49; SD = 8.88). Regarding their marital status, 53.4% reported being married, 20.6% single, 11.1% having a common-law partner, 11.6% divorced, and 3.2% separated. Out of the 189 respondents, 138 had completed university studies. In terms of job categories, the majority of the respondents were medical staff (34.9%), followed by nursing staff (28%), technical staff (23.3%), and managers (13.8%). Regarding the type of activity, 36.5% of participants worked exclusively in the emergency team, which works directly with patients, 36% in the coordination and emergency team or health intervention, 26.5% exclusively in call management or coordination, and 1.1% worked exclusively in health intervention.

The study was approved by a research ethics committee, and it conforms to the principles outlined in the Declaration of Helsinki [47]. Data were collected between January and February 2022. Participants were recruited through a short video in the organizational platform (https://www.youtube.com/watch?v=rl2JYDAxIYw (accessed on 19 May 2025)).

Participation was totally anonymous and voluntary (response rate was low, around 30%), and participants were informed of the objectives of the research before they provided consent for participation. Participants respond through a link hosted on the organization’s intranet to the questionnaire created using Unipark (v. 10.9).

### 2.2. Instruments

Sociodemographic data: Sex, age, job category and type of activity were included in the survey.

Work demands: Work demands were assessed with 9 items of the Job Content Questionnaire [48], specifically, from the psychological demands subscale (e.g., “My job requires a great mental effort”, “I don’t have enough time to do my job”). Cronbach’s alpha coefficient is acceptable (α = 0.78).

Stress: This was measured with 6 items (e.g., “I have been tired”, “I have been irritable”) from the Spanish version of the Copenhagen Psychosocial Questionnaire [49] with a five-point Likert scale. Cronbach’s alpha coefficient is excellent (α = 0.94).

Emotional exhaustion was measured with 5 items (e.g., “I feel emotionally exhausted because of my job”, “Working all day is really stressful for me”) of the Spanish version of the Maslach Burnout Inventory—General Survey (MBI-GS) [50]. The whole scale consists of three subscales (i.e., emotional exhaustion, cynicism, and professional efficacy), and its reliability ranges from α = 0.85 to α = 0.89.

Self-efficacy to cope with stressful situations. This scale by Tabernero et al. [51] assesses the capacity that professionals feel for managing stress effectively (e.g., “I feel capable of persisting in what I have proposed despite the adversities”, “I feel capable of effectively handling unexpected events”). Cronbach’s alpha coefficient for the items of this scale was α = 0.91.

Regulatory Emotional Self-Efficacy: The participants’ capability to manage their emotional life was assessed with 8 items from the Regulatory Emotional Self-Efficacy scale (RESE) [52] for negative affect, which consists of a five-point Likert scale ranging from being unable (1) to fully capable (5) (e.g., “Avoid flying off the handle when you get angry”, “Avoid getting discouraged by strong criticism”). Cronbach’s alpha coefficient was 0.91.

Healthy lifestyle: To assess the habits of healthcare emergency workers, the healthy lifestyle questionnaire was applied (EVS) [53]. It consists of a five-point Likert scale ranging from totally disagree to totally agree (e.g., “I feel good when I smoke”, “I sleep enough hours for my body to be rested”, “Normally, I eat vegetables and fruits everyday”). The twelve items of the scale are grouped into the following factors: tobacco use (α = 0.85), rest habits (α = 0.71), respecting mealtimes (α = 0.71), and balanced diet (α = 0.75).

### 2.3. Statistical Analyses

To test the serial mediating and moderating hypotheses, three different mediating models were tested by using the sixth model of the Process for SPSS (version 29) macro, with 10.000 bootstrap resamples and a confidence interval of 95%. For hypotheses 1 to 3 (mediational hypotheses), in the 3 models explored, emotional exhaustion was introduced as the dependent variable (DV), emotional regulation as the first serial mediating variable, and stress as the second serial mediating variable. The independent variables (IVs) introduced were work demands in the first model explored, self-efficacy to cope with stress in the second model explored, and healthy lifestyle in the third model explored, to respond to the first, second, and third hypotheses, respectively. Finally, for hypothesis 4, about the moderating role of healthy lifestyle in the relationship between stress and emotional exhaustion, the test for interaction between healthy lifestyle and stress was included in the third mediating model explored. Moreover, to test the hypothesized model on emotional exhaustion, with all the relationships together, an SEM model was explored with Mplus 8 [54] (Statistical power analysis for the study model was conducted using G*Power (version 3.1.9.7) by Faul et al. (2007) [55]. We used a priori computing of the required sample size for an alpha error = 0.05 and effect size R^2^ = 0.50. The result indicated that the required sample size for the model, including 13 t tests of linear multiple regression, was N = 169.) The fit of the model was analyzed using the chi-square test and its p probability level, and the RMSEA (root mean squared error of approximation), CFI (comparative fit index), and TLI (Tucker–Lewis index).

## 3. Results

### 3.1. Preliminary Analyses

Table 1 shows the correlations among the variables of the study. Results revealed significant intercorrelations in the predicted directions, providing preliminary evidence of the hypotheses of this study. Higher work demands were strongly associated with higher levels of stress, and the latter resulted strongly associated with emotional exhaustion. Moreover, having a healthy lifestyle was moderately associated with lower stress, higher emotion regulation, and lower emotional exhaustion. Another notable aspect was that significant sex differences were found in the correlations with stress, emotion regulation, and emotional exhaustion.

### 3.2. Mediation and Moderation Hypotheses

The results of the mediation analyses (Table 2) revealed that emotion regulation (first) and stress (second) mediated the relationships (a) between work demands and emotional exhaustion; (b) between self-efficacy to cope with stress and emotional exhaustion; and (c) between healthy lifestyle and emotional exhaustion, confirming H1, H2, and H3, respectively. Moreover, when performing the serial mediation analysis with healthy lifestyle as the predictor (Model 3), the test for interaction between healthy lifestyle and stress on emotional exhaustion was significant (F (1, 184) = 5.719, *p* = 0.018), confirming H4 about the moderating role of healthy lifestyle on the relationship between stress and emotional exhaustion. The moderating role of healthy lifestyle can be observed in Figure 2.

### 3.3. Predictive Model

To test the relationship between antecedents (work demands and self-efficacy to cope with stress), mediators (stress, emotion regulation), the moderator (healthy lifestyle), and the outcome (emotional exhaustion) simultaneously, we conducted a Structural Equation Modeling (SEM) analysis with Mplus. Firstly, we carried out a Test of Normality (Kolmogorov–Smirnov); however, only one factor shows a normal distribution (work demands). Therefore, we used the estimator MLR (correlation for non-normality) with MPlus. The Model Fit information with and without this correction factor is similar and the valence and power of all the relationships among factors remained similar (see Table 3, Model Fit information with/without correction factor MLR). Figure 3 shows the path diagram, which has three exogenous variables (work demands, self-efficacy to cope with stress, and healthy lifestyle) and three endogenous variables (stress, emotion regulation, and emotional exhaustion). The following indices were used to assess the fit of the model: Chi-Square Statistic (χ^2^); Root Mean Square Error of Approximation (RMSEA); the comparative fix index (CFI); and the Tucker–Lewis index (TLI). The proposed model fit the data reasonably well with χ^2^ = 2.648 (*p* = 0.45), RMSEA = 0.000, CFI = 1.000, and TLI = 1.006, with all fit indices above 0.97 and RMSEA below 0.05 [56].

The results of the SEM analysis confirmed the previous findings in a model that integrated all the variables. Work demands are related to emotional exhaustion not only directly, with a positive relationship between themselves, but also indirectly through the serial effect that work demands have on emotion regulation first, and in turn on stress. Moreover, self-efficacy to cope with stress showed an indirect effect on emotional exhaustion through emotion regulation and stress. Finally, healthy lifestyle has an indirect effect through emotion regulation and stress, and it has a moderating role between stress and emotional exhaustion. Figure 2 shows a depiction of this interaction. The slope between low stress and high stress was significant for both high healthy lifestyle (t = 4.90, *p* < 0.001) and low healthy lifestyle (t = 7.88, *p* < 0.001). When comparing both slopes and their t-values, it can be concluded that a less healthy lifestyle can increase susceptibility to stress and aggravate levels of emotional exhaustion, while a healthy lifestyle can have a mitigating or protective effect by reducing the influence of stress on emotional exhaustion.

Moreover, a statistically significant difference was found in the data when higher levels of stress were reported. Specifically, emotional exhaustion turned out lower when there was a high healthy lifestyle than when there was a low healthy lifestyle (t = −2.60, *p* = 0.009). This significant difference was not found when lower levels of stress were reported (t = 0.38, *p* = 0.702).

When controlling for the effect of sex on stress in the SEM model, we found a statistically significant positive relationship (β = 0.13, *p* < 0.01). Additionally, when controlling for the effect of sex on emotion regulation, we found a statistically significant negative relationship (β = −0.15, *p* < 0.01). This indicates that there are discernible differences in stress levels and emotion regulation strategies among men and women. However, when controlling for the effect of sex on emotional exhaustion, the relationship was not significant (β = 0.03, n.s.), which indicates that sex did not play a role in explaining the variability in emotional exhaustion perception.

## 4. Discussion

Healthcare professionals have shown to be particularly vulnerable to burnout, which is why several studies have identified organizational risk factors that contribute to its development, such as high work demands, long working hours, high or frequent rotations of overtime, and individual factors such as job experience, personality traits, coping styles, and emotion regulation skills [57]. Burnout is known to have serious implications on their job performance, reducing empathy [58], affecting the quality of patient treatment [59], and increasing rates of major medical errors [60].

Although burnout was already a significant concern for healthcare professionals, the COVID-19 pandemic led to an increase in their vulnerability [61], which is why, based on the JD-R model, this study explored how work demands and personal resources, such as self-efficacy to cope with stress, emotion regulation, and healthy habits, interact and associate with the main component of burnout: emotional exhaustion. Specifically, the purpose of this study was to identify predictors, mediators, and moderators that have an impact on emotional exhaustion in emergency healthcare workers.

The results of this research provide support for H1, which predicted that emotion regulation and stress would act as serial mediators in the relationship established between work demands and emotional exhaustion. Moreover, the serial mediating role of emotion regulation and stress in the relationship established between self-efficacy to cope with stress and emotional exhaustion expected in H2 was also supported. Additionally, emotion regulation and stress were shown to be significant mediators in the relationship between healthy lifestyle and emotional exhaustion, as stated in H3. Finally, healthy lifestyle moderated the strength of the mediated relationship between stress and emotional exhaustion, supporting H4.

Our findings are consistent with previous research. In general, studies have shown that emotional exhaustion is positively related to job demands, such as work overload and emotional demands, while being negatively related to job resources [62]. Additionally, emotion regulation skills have been found to play a fundamental role in how healthcare workers cope with workplace stressors, influencing their overall well-being and ultimately their practice [34,63]. Although self-efficacy to cope with stress has been little studied, especially in the emergency healthcare context, it is known that low levels of general self-efficacy led to stress and burnout [64,65]. Regarding healthy lifestyle, poor sleep quality as a result of workload has been found to relate to emotional exhaustion and struggles when recovering from stress [66]. In this sense, healthy habits and their promotion have been essential for the development of intervention programs for preventing burnout and substance use disorder in healthcare professionals [67].

Our results provide empirical support for the JD-R model in the context of emergency healthcare, introducing personal resources as a process that impacts on the development of emotional exhaustion, thus providing relevant relationships that make job resources more effective (see Figure 4). Following Bakker et al. [22], personal resources are known to moderate the impact of job demands on employee well-being. In this sense, the identification of healthy lifestyle as a significant moderator in the mediated relationship between stress and emotional exhaustion contributes to understanding how recovery-enhancing behaviors beyond personal resources can mitigate the negative effects of work demands.

This thesis contributes to the JD-R literature by integrating healthy lifestyle habits as an additional layer of resources within the healthcare sector. Whereas the JD-R model has traditionally emphasized organizational and personal resources (e.g., autonomy, social support, resilience, optimism) to explain the dynamics of burnout and engagement [22,26], this work argues that lifestyle habits—such as exercise, sleep quality, and diet—constitute recovery-enhancing behaviors that play a complementary, buffering role. From a JD-R perspective, these habits do not replace personal resources but function as external moderators, helping healthcare workers to replenish depleted energy, sustain engagement, and protect health in the face of high emotional and cognitive demands [45]. Recognizing the importance of lifestyle habits thus broadens the explanatory scope of the JD-R framework and offers a more holistic understanding of resilience in healthcare. Importantly, this perspective highlights that investing in health-promoting behaviors has implications not only for staff well-being but also for patient safety and quality of care, thereby linking individual recovery to organizational and societal outcomes [46].

Taken together, the interaction between personal resources and job demands has practical implications for emergency healthcare workers. Firstly, the importance of incorporating self-care, both emotional and physical, in coping with stress and preventing burnout cannot be overstated. Encouraging workers to adopt healthy habits such as regular exercise, a balanced diet, and enough rest can help to reduce the effect of stress on emotional exhaustion. Secondly, training on stress management techniques and emotion regulation can help professionals to cope with such a demanding setting as the emergency healthcare sector [68].

One limitation of the present study is that the data were gathered in a period where COVID-19 cases were rising. The unpredictability of COVID-19 and its unknown extent may have increased psychological stress [69], which is not related to work demands and could have acted as an extraneous variable. Additionally, a key limitation of the present study is its cross-sectional design, which precludes drawing conclusions about the causal direction of the observed associations. Although the findings are consistent with the JD-R framework, it is not possible to determine whether job demands and resources influence stress and emotional exhaustion, or whether these outcomes in turn shape perceptions of work demands and personal resources. Consequently, the potential for reciprocal or bidirectional effects cannot be ruled out. We suggest that future research aims for a longitudinal design to make comparisons over time. Another limitation is the potential for sampling bias due to voluntary participation, as individuals experiencing higher or lower levels of stress and well-being may have been more inclined to take part, which could affect the representativeness of the findings.

Although personal resources are essential to prevent emotional exhaustion in the healthcare context, job resources such as workplace support have shown a significant role in burnout prevention [37]. Furthermore, variables such as gender and age could be considered for a more extensive comprehension of emotional exhaustion, since older professionals have shown to be more engaged with their work, use more emotion regulation strategies, and be more confident with their abilities, thus experiencing less emotional exhaustion in comparison to younger employees [70], and women appear to be more affected by job demands and psychological distress than men [71].

It is important to acknowledge that gender differences may have both theoretical and practical implications for how the JD-R model is applied in the healthcare sector. For example, women may report higher emotional demands and use different coping strategies to men, while men may perceive distinct resources as more salient. Such variations suggest that the pathways linking demands, resources, and well-being outcomes could operate differently across genders, raising the need for more gender-sensitive formulations of the JD-R framework. Addressing these differences would not only enrich the theoretical understanding of stress and emotion regulation but also inform the design of tailored interventions to support healthcare workers more effectively.

Moreover, regarding age, our sample suggests a bias toward older healthcare workers. On the one hand, this may limit the generalizability of the findings to younger professionals, who might experience job demands, resources, and well-being outcomes differently at earlier career stages. On the other hand, the age distribution may also reflect the demographic reality of the healthcare workforce in many countries, where aging staff are increasingly common, meaning that the sample could still provide a realistic picture of the sector. Importantly, this highlights the need to consider age as a potential moderator in JD-R analyses, since age-related differences in resilience, recovery strategies, and career expectations could shape how demands and resources influence outcomes [25].

Nevertheless, this research contributes to the study of the variables that influence the development of emotional exhaustion in emergency healthcare workers by highlighting the importance of individual factors. It further implicates that healthcare organizations should implement interventions that promote healthy habits, emotion regulation skills and coping strategies to deal with high work demands in order to prevent the development of burnout and its consequences. Finally, this study suggests the incorporation of the aforementioned personal resources into future research that aims to provide an extensive overview of occupational stress and burnout in emergency healthcare professionals.

## 5. Conclusions

This cross-sectional study with emergency healthcare workers in Spain tested a JD-R model and showed that personal resources and lifestyle meaningfully shape emotional exhaustion. Emotion regulation and stress operated as sequential mediators linking (a) work demands, (b) self-efficacy to cope with stress, and (c) healthy lifestyle to emotional exhaustion. In addition, healthy lifestyle buffered the stress–exhaustion relationship, indicating that recovery-enhancing behaviors (sleep, nutrition, and physical activity) can mitigate the impact of stress on burnout. Gender differences in stress and emotion regulation (but not in emotional exhaustion) point to the value of gender-sensitive approaches in organizational interventions.

Taken together, the findings extend the JD-R framework by positioning healthy lifestyle as a complementary, external moderator that works alongside psychological resources (e.g., emotion regulation and stress-coping self-efficacy). Practically, this implies that multi-level prevention is needed in emergency services, namely (1) individual-level programs that build emotion-regulation and stress-management skills and strengthen self-efficacy, and (2) organizational measures that protect basic recovery (e.g., scheduling that protects sleep and breaks, access to nutritious food, opportunities for physical activity).

Future research should use longitudinal designs, combine subjective and objective indicators of stress and recovery, and test targeted interventions that integrate lifestyle promotion with enhancements to job resources. Examining potential moderators such as gender and age will further clarify for whom and under what conditions these mechanisms most effectively reduce emotional exhaustion in emergency healthcare.

## Figures and Tables

**Figure 1 healthcare-13-02336-f001:**
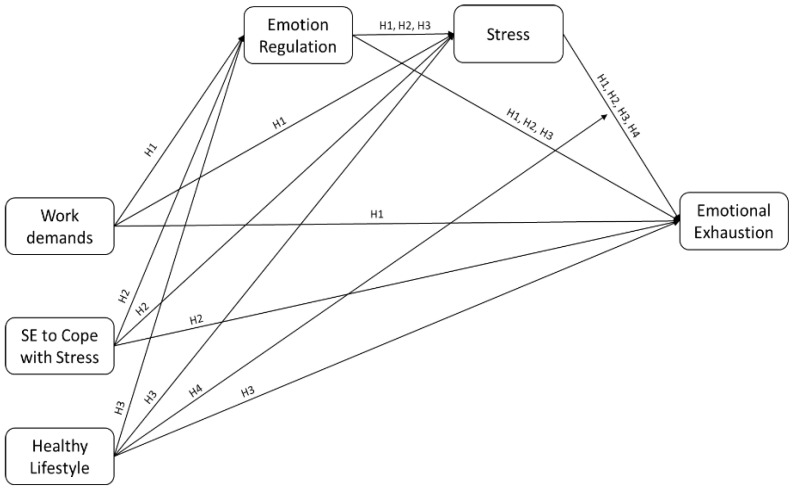
Hypothesized model of emotional exhaustion.

**Figure 2 healthcare-13-02336-f002:**
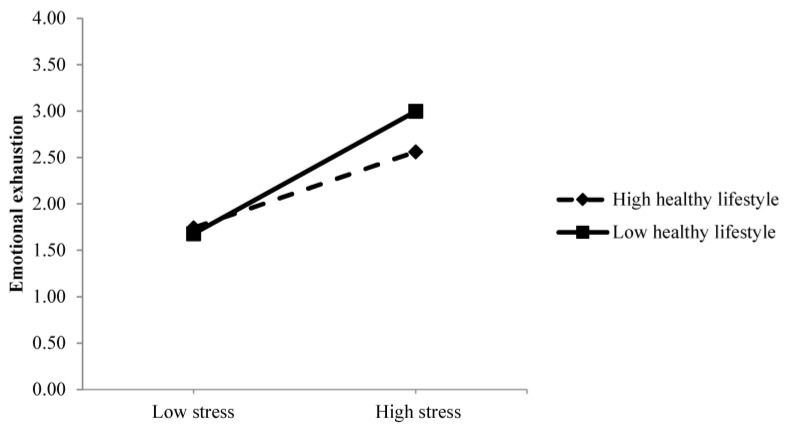
Interaction between stress and healthy lifestyle in relation to emotional exhaustion.

**Figure 3 healthcare-13-02336-f003:**
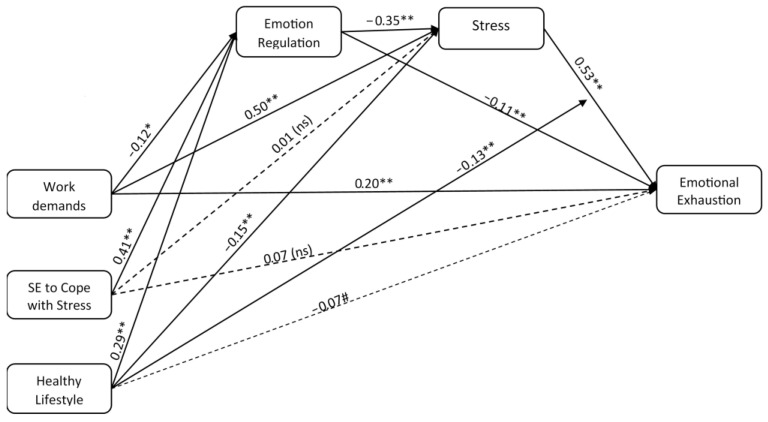
Results of Structural Equation Modeling with Mplus. Note: ** *p* < 0.01; * *p* < 0.05; # *p* < 0.10.

**Figure 4 healthcare-13-02336-f004:**
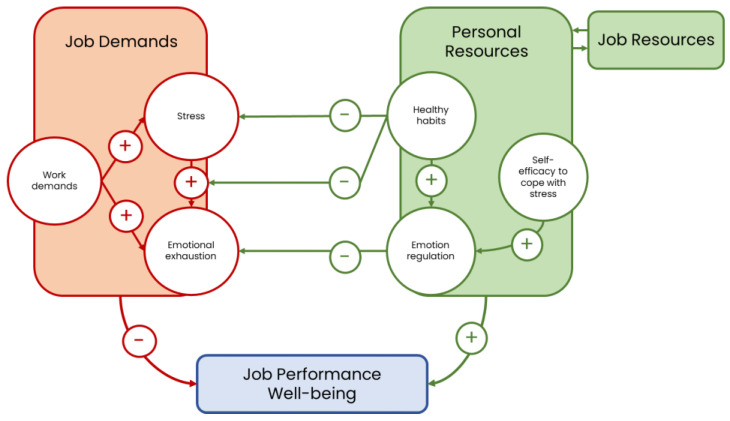
Depiction of the results of this study in the frame of the JD-R model.

**Table 1 healthcare-13-02336-t001:** Mean, standard deviations, and correlations of study variables.

	M	SD	1	2	3	4	5	6
1. Sex (0 = male; 1= female)			-					
2. Work demands	3.07	0.68	0.14	-				
3. Stress	2.81	0.95	0.30 **	0.63 **	-			
4. Self-efficacy to cope with stress	4.27	0.67	0.12	−0.10	−0.27 **	-		
5. Emotion regulation	3.59	0.86	−0.25 **	−0.24 **	−0.57 **	0.51 **	-	
6. Healthy lifestyle	3.21	0.57	−0.10	−0.19 *	−0.40 **	0.23 **	0.42 **	-
7. Emotional exhaustion	2.25	1.17	0.25 **	0.58 **	0.75 **	−0.19 **	−0.49 **	−0.40 **

Note. * *p* < 0.05 (2-tailed); ** *p* < 0.01 (2-tailed).

**Table 2 healthcare-13-02336-t002:** Results of the mediation analyses of emotion regulation and stress as mediators in the relationships between work demands and emotional exhaustion (H1), between self-efficacy to cope with stress and emotional exhaustion (H2), and between healthy lifestyle and emotional exhaustion (H3).

	Emotion Regulation (M1)		Stress (M2)		Emotional Exhaustion (Y)	
Model 1 (H1)	Coeff.	SE	Coeff.	SE	Coeff.	SE
Work demands (X)	−0.24 ***	0.09	0.52 ***	0.07	0.20 **	0.11
Emotion regulation (M1)	-	-	−0.44 ***	0.05	−0.12 *	0.08
Stress (M2)	-	-	-	-	0.56 ***	0.09
Model settings	R^2^ = 0.06 F (1, 187) = 11.429 ***		R^2^ = 0.58 F (2, 186) = 126.379 ***		R^2^ = 0.59 F (3, 185) = 88.673 ***	
Indir. cond. effect Bootstrap (95% CI)	X → M1 → Y 0.030 [0.001, 0.068]		X → M2 → Y 0.291 [0.201, 0.387]		X → M1 → M2 → Y 0.059 [0.023, 0.101]	
**Model 2 (H2)**						
Self-efficacy (X)	0.51 ***	0.08	0.02 (ns)	0.10	0.06 (ns)	0.10
Emotion regulation (M1)	-	-	−0.57 ***	0.08	−0.12 #	0.09
Stress (M2)	-	-	-	-	0.70 ***	0.07
Model settings	R^2^ = 0.26 F (1, 187) = 64.740 ***		R^2^ = 0.32 F (2, 186) = 43.572 ***		R^2^ = 0.57 F (3, 185) = 81.784 ***	
Indir. cond. effect Bootstrap (95% CI)	X → M1 → Y −0.063 [−0.155, 0.008]		X → M2 → Y 0.014 [−0.087, 0.120]		X → M1 → M2 → Y −0.203 [−0.303, −0.114]	
**Model 3 (H3)**						
Healthy lifestyle (X)	0.42 ***	0.10	−0.20 **	0.11	−0.10 #	0.11
Emotion regulation (M1)	-	-	−0.48 ***	0.07	−0.06 (ns)	0.08
Stress (M2)	-	-	-	-	0.67 ***	0.07
Model settings	R^2^ = 0.18 F (1, 187) = 40.120 ***		R^2^ = 0.35 F (2, 186) = 50.378 ***		R^2^ = 0.57 F (3, 185) = 83.329 ***	
Indir. cond. effect Bootstrap (95% CI)	X → M1 → Y −0.027 [−0.081, 0.030]		X → M2 → Y −0.134 [−0.228, −0.038]		X → M1 → M2 → Y −0.136 [−0.210, −0.079]	

Notes. X = independent variable; M = mediator; Y = dependent variable; Indir. cond. effect = indirect conditional effect; CI = confidence interval; Coeff. = coefficient; SE = standard error. *** *p* < 0.001; ** *p* < 0.01; * *p* < 0.05; # *p* < 0.10.

**Table 3 healthcare-13-02336-t003:** Model fit values with and without the correction factor for the estimator MLR (correlation for non-normality) with MPlus.

	Without Correction Factor		With Correction Factor for MLR	
**Chi-square**	Value	3.658	Value	2.648
	Degrees of Freedom	3	Degrees of Freedom	3
	*p*-Value	0.3009	*p*-Value	0.4492
			Scaling Correction Factor for MLR	1.3814
**RMSEA**	Estimate	0.034	Estimate	0.000
	90 Percent C.I.	0.000–0.132	90 Percent C.I.	0.000–0.117
	Probability RMSEA ≤ 0.05	0.492	Probability RMSEA ≤ 0.05	0.632
**CFI/TLI**	CFI	0.998	CFI	1.000
	TLI	0.991	TLI	1.006

## Data Availability

The data presented in this study are available on request from the corresponding author due to ethical reasons.

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
