# Peer review of "Understanding the Impact of Personal Resources on Emotional Exhaustion Among Emergency Healthcare Workers: A Structural Equation Modeling Approach"

_healthcare, 2025, doi:10.3390/healthcare13182336_

Round 1

Reviewer 1 Report

Comments and Suggestions for Authors

The paper explores an important topic using the Job Demands-Resources (JD-R) model. It employs appropriate statistical techniques (SEM, PROCESS macro) to investigate emotional exhaustion among emergency healthcare workers. The study contributes by integrating healthy lifestyle and self-efficacy into the model. However, there are significant areas that need revision to strengthen the paper's theoretical framing, clarity, and academic rigor.

  1. The manuscript uses the JD-R model but does not clearly state what gap in the literature it fills. Please clearly articulate how this study offers a novel contribution. Frame this explicitly in both the Introduction and Discussion sections.
  2. The term “personal resources” is used inconsistently, at times equated with healthy habits and at other times with self-efficacy and emotion regulation. The authors need to define "personal resources" early with reference to JD-R theory and clarify whether lifestyle habits fit this definition or are an external, moderating buffer.

  3. The limitations of a cross-sectional design are acknowledged only briefly. The authors need to expand this section. Emphasize that causality cannot be inferred and that reciprocal effects cannot be ruled out.

  4. The manuscript briefly mentions mindfulness as an emotion regulation technique associated with reduced burnout (p. 3, line 102), but this is not supported with sufficient references. Given the relevance of mindfulness-based interventions in stress and burnout reduction, I strongly recommend the authors add foundational references to support this statement.

    - The Contributions of Mindfulness Meditation on Burnout, Coping Strategy, and Job Satisfaction: Evidence from Thailand, Journal of Management and Organization, 19(5), 544-558. doi.org/10.1017/jmo.2014.8

    - The Benefits of Mindfulness Meditation: Changes in Emotional States of Depression, Anxiety, and Stress. Behaviour Change, 25(3), 156–168. doi:10.1375/bech.25.3.156

  5. Clearly state the sampling method (e.g., convenience, purposive, voluntary response) and discuss sampling bias in the limitations section.

  6. No information is provided on which regions of Spain or how many organizations/hospitals participated. Without this, it's unclear if the sample is geographically or organizationally representative.

  7. The sample has a mean age of 49 with an SD of 8.88, suggesting a bias toward older healthcare workers. Discuss whether this could be the problem.

  8. Gender effects were found in stress and emotion regulation but were not meaningfully discussed. It is important to discuss theoretical implications of these gender differences.

Reviewer 2 Report

Comments and Suggestions for Authors

Thank you for the opportunity to review this manuscript. Please see my feedback below:

General comments:

  • A paragraph should be at least 3 sentences long

Introduction:

  • It would be helpful to describe and critically analyse prior research that has applied the JD-R model in the healthcare sector
  • Some headings would improve the flow of the introduction e.g., burnout, JD-R model, personal resources, the current study

Method:

  • How were participants recruited?
  • How many people worked in at the hospital/s you recruited from? An indication of response rate would be helpful.
  • Were participants recruited during the pandemic? There is mention of the pandemic in the introduction and discussion but not in the method
  • Are the reliability coefficients provided from prior research or the current study?
  • Unfinished sentence line 196-197 about RESE
  • Do you have a power calculation to assert that your sample size was sufficient for the conducted analyses?

Results:

  • What kind of correlations are reported in Table 1? Pearson’s correlations should not be conducted on categorical/ dichotomous variables (e.g., sex).
  • Did you conduct any assumption testing and what were the outcomes?

Round 2

Reviewer 1 Report

Comments and Suggestions for Authors

All the requested revisions have been addressed thoroughly. The manuscript is now clear, well-structured, and suitable for publication.

Author Response

Thank you for your helpful comments.

Reviewer 2 Report

Comments and Suggestions for Authors

Thank you for your revisions, the manuscript is much improved!

I still believe further critical analysis could be incorporated into the introduction. The text added was helpful but largely just described the JDR model. Please also check this section for clarity and flow.

Please include the power calculation in the manuscript.

Thanks!
